# C-Reactive Protein-to-Prealbumin and C-Reactive Protein-to-Albumin Ratios as Nutritional and Prognostic Markers in Hospitalized Patients—An Observational Study

**DOI:** 10.3390/nu16162610

**Published:** 2024-08-08

**Authors:** Rosa M. García-Moreno, Laura Mola Reyes, Bricia López-Plaza, Samara Palma Milla

**Affiliations:** 1Clinical Nutrition and Dietetics Unit, Department of Endocrinology and Nutrition, Hospital Universitario La Paz, Paseo de la Castellana, No. 261, 28046 Madrid, Spain; lauramolareyes@gmail.com (L.M.R.); palmamillasamara@gmail.com (S.P.M.); 2Food, Nutrition and Health Platform, Hospital La Paz Institute for Health Research (IdiPAZ), 28046 Madrid, Spain; bricia.plaza@idipaz.es

**Keywords:** C-reactive protein, prealbumin, albumin, malnutrition, mortality, hospitalization

## Abstract

This study aimed to evaluate the role of the C-reactive protein-to-prealbumin (CP) ratio and the C-reactive protein-to-albumin (CA) ratio as nutritional and prognostic markers. A retrospective study was conducted on hospitalized patients who underwent a nutritional assessment and the measurement of C-reactive protein, prealbumin, and albumin (*n* = 274). Receiver operating characteristic (ROC) curve analysis was used. The area under the curve (AUC) of the CP ratio for predicting mortality was 0.644, 95%CI (0.571 to 0.717), and the CA ratio had an AUC of 0.593, 95%CI (0.518 to 0.669). The AUC of the CP ratio for the differential diagnosis between severe and moderate disease-related malnutrition (DRM) was 0.643, 95%CI (0.564 to 0.722), and the CA ratio had an AUC of 0.650, 95%CI (0.572 to 0.728). The CP and CA ratios showed greater accuracy in predicting mortality compared to C-reactive protein as an isolated marker (*p* = 0.011 and *p* = 0.006, respectively). Both ratios also improved the ability of prealbumin and albumin to identify severe DRM (*p* = 0.002 and *p* = 0.044, respectively). In conclusion, these results suggest that the CP and CA ratios may have a limited role in predicting mortality and identifying severe DRM by outperforming isolated protein markers.

## 1. Introduction

Disease-related malnutrition (DRM) is a complex syndrome resulting from inadequate nutrient intake that does not fulfill the patient’s physiological requirement and disease-related systemic inflammatory response [1]. DRM is a common problem, with an estimated prevalence of almost 30% of hospitalized patients in Spain [2] and a similar or even higher prevalence in other countries [3,4,5]. There is a strong association between DRM and increased risk of morbidity, mortality, prolonged hospital stay, higher rates of readmission, and increased costs [1,2].

Serum visceral proteins have been classically used in routine clinical practice to facilitate the detection of DRM and monitoring refeeding in malnourished patients. Albumin and prealbumin are serum visceral proteins synthesized in the liver and widely evaluated as nutritional and prognostic markers. However, both are non-specific markers for nutritional status since they can be modified in response to diverse conditions such as liver disease or inflammation. Because of the reprioritization of liver protein synthesis during inflammatory processes, serum visceral proteins decrease proportionally to the elevation in acute-phase proteins, such as C-reactive protein (CRP) [6]. CRP is an acute-phase protein synthesized by hepatocytes and broadly used as a non-specific diagnostic marker of inflammation [7,8]. It has been proposed that prealbumin is not interpretable as a nutritional marker when the CRP level is higher than 15 mg/dL, while if the CPR level is lower, prealbumin allows for the detection of malnutrition and the degree of malnutrition [9]. 

Inflammation is one of the etiological factors that most contributes to the appearance of DRM, with it being present in a high proportion of malnourished patients. Since prealbumin and albumin as isolated markers lose value in patients with inflammation, other laboratory markers have been proposed. Thus, different parameters that relate inflammatory and nutritional markers have emerged, such as CRP-to-prealbumin (CP) ratio or CRP-to-albumin (CA) ratio [7,10,11]. However, their validity as prognostic markers, optimal cut-off points, and potential role in the diagnosis of DRM and the degree of malnutrition are still to be determined. 

This study aimed to evaluate the role of the CP ratio and the CA ratio as prognostic markers of mortality, complications, hospital length of stay (LOS), and readmissions, as well as the correlation of these biomarkers with the diagnosis of DRM in hospitalized patients.

## 2. Materials and Methods

### 2.1. Study Design and Subjects

This retrospective, single-center study was conducted in a tertiary hospital between April and September 2022. The participants were selected among the hospitalized patients evaluated by the Clinical Nutrition Unit staff in routine clinical practice. Among a total of 380 patients assessed within the study period, 274 subjects met the selection criteria and were included. The inclusion criteria were: hospitalized patients older than 18 years who underwent nutritional screening, nutritional assessment, and diagnosis; measurement of serum CRP, prealbumin, and albumin at the time of the first nutritional assessment; and life expectancy longer than three months. Exclusion criteria were the lack of any of these clinical or laboratory data. 

### 2.2. Nutritional Assessment

The nutritional assessment of the patients was carried out by trained medical personnel from the nutrition unit. The CONtrolling NUTritional status (CONUT) score was used for nutritional screening [12], and the diagnosis of malnutrition was performed using both the Subjective Global Assessment (SGA) and the Global Leadership Initiative on Malnutrition (GLIM) criteria [13,14]. The criterion of low muscle mass was evaluated via anthropometry using mid-arm muscle circumference (MAMC) or calf circumference (CC) measurements. Inflammation was defined as CRP values above the upper limit of normal (ULN) established by our laboratory (CRP > 5 mg/L). 

### 2.3. Biochemical Markers 

The blood samples were collected first thing in the morning, and the recommended procedures for collecting diagnostic blood specimens via venipuncture were followed [15]. For measurement, blood samples were allowed to clot completely before centrifugation, and the tubes were kept capped at all times. Measurements were not performed in triplicate since the research was conducted in routine clinical practice. The determination of the serum biochemical markers was carried out in our hospital laboratory. All measurements were performed using an Atellica^®^ CH Analyzer (Siemens Healthcare GmbH, Erlangen, Germany).

The albumin assay was based on the method of Doumas and Biggs that uses bromocresol green solution (BCG) as a binding dye [16]. The serum albumin quantitatively binds to BCG to form an albumin–BCG complex that is measured as an endpoint reaction. The system automatically performs the following steps. Firstly, 50 µL of the primary serum sample and 200 µL of Atellica CH diluent were dispensed into a dilution cuvette. After that, 20 µL of reagent A (bromocresol green (1.0 mmol/L)) and 80 µL of special reagent water were added into a reaction cuvette. Subsequently, 2 µL of pre-diluted sample was dispensed into the reaction cuvette. The mixture was incubated at 37 °C. Finally, after reagent A addition, the absorbance was measured at wavelengths of 596/694 nm. 

Prealbumin was determined using the turbidimetric assay [17]. The system automatically quantifies prealbumin by performing the following procedure. Firstly, 50 μL of the primary serum sample and 200 μL of Atellica CH Diluent were dispensed into a dilution cuvette. Subsequently, 75 μL of reagent A (polyethylene glycol (4%); tris-HCl buffer (20 mmol/L); sodium chloride (150 mmol/L); sodium azide (0.09%)) was dispensed into a reaction cuvette. After this, 7.5 μL of prediluted sample was dispensed to measure the absorbance. In the next step, 15 μL of reagent B (anti-human prealbumin antibodies (goat); tris-HCl buffer (20 mmol/L); sodium chloride (150 mmol/L); sodium azide (0.09%)) was dispensed, mixed, and incubated at 37 °C. Finally, absorbance was measured at the wavelength of 340 nm after adding reagent B to obtain the results. 

High-sensitivity CRP was measured in serum using a latex-enhanced immunoturbidimetric assay [18]. It is based on the principle that the analyte concentration is a function of the intensity of scattered light caused by the latex aggregates. The latex particles coated with anti-CRP antibodies rapidly agglutinate in the presence of CRP. The Atellica CH Analyzer automatically performs the following steps. Firstly, 50 µL of the primary plasma sample and 200 µL of Atellica CH Diluent were dispensed into a dilution cuvette. After that, 50 µL of reagent A (glycine (170 mmol/L); sodium chloride (100 mmol/L); EDTA disodium salt dihydrate (50 mmol/L); sodium azide (0.09%)) was added into a reaction cuvette. Subsequently, 10 µL of pre-diluted sample was dispensed into the reaction cuvette. Afterward, 50 µL of reagent B (CRP antibody synthetic latex (rabbit); sodium azide (0.09%)) was dispensed into the reaction cuvette. This mixture was incubated at 37 °C. Finally, the absorbance was measured at wavelengths of 571/658 nm after adding reagent B. 

The laboratory data were collected in the units of measurement provided in the laboratory reports of our center (CPR was measured in mg/L, prealbumin was measured in mg/dL, and albumin was measured in g/dL). All of these parameters were converted to the same units (mg/dL) to calculate both the CP ratio and the CA ratio. 

### 2.4. Variables

Data on the age, gender, and medical history of the patients, cause of hospital admission, nutritional screening and diagnosis, medical nutrition therapy, biochemical markers, and outcomes were collected.

The outcomes evaluated were the diagnosis of DRM, the development of infectious or non-infectious complications, LOS, mortality during admission or, for those patients transferred to a continuity of care center, mortality during the three subsequent months after transfer, and readmission in the three months after discharge.

The laboratory biomarkers of interest were the CP ratio and the CA ratio, studying their association with the outcomes. These biomarkers were measured in all patients in the first nutritional assessment, and a second determination of these ratios was collected coinciding with the last nutritional assessment in those patients whose LOS and follow-up by the Nutrition staff exceeded ten days. 

### 2.5. Statistical Analysis

For descriptive analysis, the categorical data were reported in absolute value and percentage. The continuous data were described in mean and standard deviation (SD) for parametric variables, or median and interquartile range (IQR) for non-parametric variables. The Kolmogorov–Smirnov test was used to evaluate whether the continuous variables followed a parametric distribution. Receiver operating characteristics (ROC) curves were generated for the CP and CA ratios to estimate their accuracy as markers of the outcomes studied. The areas under the curve (AUC) of the ROC curves were compared using the bootstrap method. The optimal cut-off points of the CP and CA ratios were calculated using the Youden index method. The sensitivity, specificity, and positive and negative predictive values of the cut-off points were estimated. Spearman’s correlation coefficient (Rho) was used to assess the correlation between non-parametric continuous data. The CP and CA ratios were categorized according to their optimal cut-off points to predict mortality, and subgroup analyses were performed. The chi-square (χ^2^) statistic was used for categorical data, and the *t*-Student test and the Mann–Whitney U test were used for continuous parametric and non-parametric independent data, respectively. The Wilcoxon Signed-Rank test was used to compare paired non-parametric data. Logistic regression analysis was performed to estimate the effect of the CP and the CA ratios above their optimal cut-off points on the risk of mortality, and the results were expressed in odds ratios (OR). Stratified analyses were conducted to correct any potential bias induced by the heterogeneity regarding the cause of admission to hospital of patients. In all analyses, a *p*-value < 0.05 was considered statistically significant. The statistical software used was R version 4.1.2 (R Foundation for Statistical Computing, Vienna, Austria).

## 3. Results

### 3.1. Description of Results

A total of 274 patients were included (37.6% women) with an average age of 66 ± 17 years. Approximately 69% of patients had DRM according to GLIM criteria and 85% had inflammation. The main characteristics of the patients are described in Table 1. The median CP ratio at the first assessment was 0.295 (IQR 0.817) and the CA ratio was 0.115 (IQR 0.010). 

### 3.2. ROC Curve Analysis

The AUC of ROC curves of the CP ratio and the CA ratio in the first assessment are shown in Table 2. ROC curves for mortality, complications, readmissions, and nutritional diagnosis according to SGA and GLIM criteria were generated.

The CP ratio showed no correlation with LOS (Rho 0.036, *p* = 0.556), as well as the CA ratio (Rho 0.069, *p* = 0.253).

#### 3.2.1. Evaluation of the CP Ratio and the CA Ratio as Predictors of Mortality

The ROC curves of the CP ratio, CA ratio, CRP, prealbumin, and albumin for mortality are represented in Figure 1A. The CP ratio had higher accuracy in predicting mortality than the CA ratio (*p* = 0.029) and CRP (*p* = 0.011), but without significant differences with prealbumin (*p* = 0.758) and albumin (*p* = 0.590). The CA ratio was a better-quality prognostic marker for mortality compared to CRP (*p* = 0.006), with no differences compared to prealbumin (*p* = 0.219) and albumin (*p* = 0.067). 

The optimal cut-off point of the CP ratio to predict mortality was 0.153. The sensitivity, specificity, positive predictive value (PPV), and negative predictive value (NPV) of a CP ratio ≥ 0.153 to predict mortality are described in Table 3. In univariate logistic regression, a CP ratio ≥ 0.153 was a predictor of increased risk of mortality (OR 5.23; 95%CI (2.39 to 13.17); *p* < 0.001). In a multivariate logistic regression model adjusted by age, medical history of diabetes mellitus (DM), hypertension, dyslipidemia, obesity, ischemic cardiomyopathy, chronic kidney disease (CKD), liver disease, chronic obstructive pulmonary disease (COPD), obstructive sleep apnea (OSA) and cancer, cause of admission, nutritional diagnosis according to GLIM criteria, SGA, and inflammation, the statistical significance of CP ratio ≥ 0.153 was maintained (OR 7.98; 95%CI (2.85 to 26.89), *p* < 0.001).

The optimal cut-off point of the CA ratio to predict mortality was 0.040. The prognostic accuracy of the CA ratio ≥ 0.040 for mortality prediction is shown in Table 3. Logistic regression showed that a CA ratio ≥ 0.040 was a predictor for mortality (OR 7.57, 95%CI (2.66 to 31.90), *p* < 0.001). In a multivariate logistic regression model adjusted by age, medical history of DM, hypertension, dyslipidemia, obesity, ischemic cardiomyopathy, CKD, liver disease, COPD, OSA and cancer, cause of admission, diagnosis according to GLIM criteria, and SGA, the CA ratio ≥ 0.040 remained an independent predictor of mortality (OR 11.38; 95%CI (3.49 to 54.08), *p* < 0.001).

#### 3.2.2. Evaluation of the CP Ratio and the CA Ratio as Predictors of Malnutrition

Both the CP ratio and CA ratio showed poor diagnostic accuracy for malnutrition but a slightly better capacity to differentially diagnose severe to moderate malnutrition (Table 2). ROC curves of CP ratio, CA ratio, prealbumin, and albumin for the differential diagnosis between severe and moderate malnutrition according to GLIM criteria are displayed in Figure 1B. When comparing the ROC curves, the CP ratio had higher accuracy in detecting severe malnutrition than prealbumin (*p* = 0.002), and the CA ratio was also better than albumin (*p* = 0.044). The optimal cut-off point of the CP ratio and the CA ratio to detect severe malnutrition according to GLIM criteria was 0.237 for both parameters. The diagnostic accuracy of the CP ratio < 0.237 and CA ratio < 0.273 to differentiate between severe and moderate malnutrition is described in Table 3. 

#### 3.2.3. Characteristics of the Subgroups of the Patients Stratified According to the CP Ratio and the CA Ratio Values

The characteristics of the subgroups of the patients stratified according to the cut-off value of the CP ratio and the CA ratio collected in the first assessment are described in Table 1.

When comparing the patients with a CP ratio < 0.153 and those with a CP ratio ≥ 0.153, there was a higher prevalence of obesity (*p* = 0.011) in the subgroup with a CP ratio ≥ 0.153. Additionally, there was a significantly higher proportion of patients with positive CONUT nutritional screening (*p* < 0.001), as well as patients with SGA score B and moderate DRM according to GLIM criteria (*p* = 0.013 and *p* = 0.023, respectively). Logically, the prevalence of mortality was significantly higher in the subgroup of patients with a CP ratio ≥ 0.153 (*p* < 0.001), without differences in the outcomes of LOS, complications, and readmissions. 

When comparing the patients with a CA ratio < 0.040 and those with a CA ratio ≥ 0.040, it stands out that there is a significantly higher proportion of surgical patients within the subgroup with a CA ratio ≥ 0.040 (*p* = 0.011). There was also a higher proportion of patients with a moderate or high risk of malnutrition in CONUT nutritional screening in this subgroup (*p* = 0.001) but with no significant differences regarding SGA score and nutritional diagnosis according to GLIM criteria. Moreover, the patients with a CA ratio ≥ 0.040 had a higher prevalence of infectious complications and mortality (*p* = 0.045 and *p* < 0.001, respectively).

### 3.3. Evolution of the CP Ratio and the CA Ratio during Follow-Up

A second determination of the CP and CA ratios was performed in 128 patients, coinciding with the ending of the nutritional follow-up during hospitalization. The median CP ratio at the last assessment was 0.188 (IQR 0.585) and the CA ratio was 0.082 (IQR 0.192). Both the CP and CA ratios decreased significantly throughout hospitalization (*p* < 0.001 for both parameters). The evolution of the median of the CP and the CA ratios in the different subgroups of patients according to their nutritional diagnosis, nutritional therapy, and outcomes is displayed in Figure 2.

Patients who ended up dying had an increase in the CA ratio during follow-up, unlike those who survived, in whom the CA ratio tended to decrease (Figure 2B). Therefore, the evolution of this parameter was significantly different in both subgroups (*p* = 0.005). 

Regarding the CP ratio, it tended to decrease more sharply in the subgroup of patients who survived compared with those who died at the end of the study. Nevertheless, no statistically significant difference regarding the evolution of this parameter was found (*p* = 0.071) (Figure 2A).

There were no differences in the evolution of both the CP ratio and the CA ratio in the different subgroups depending on the nutritional diagnosis, with a non-significant interaction effect (*p* = 0.066 and *p* = 0.059, respectively) (Figure 2C,D). However, as may be observed, the severe DRM subgroup started from lower values of both parameters. 

The type of nutritional therapy also did not significantly influence the evolution of the CP ratio and the CA ratio during follow-up (Figure 2E,F), as shown by the non-significant interaction effect (*p* = 0.467 and *p* = 0.068, respectively). Notably, the decrease in both the CP and CA ratios tended to be more pronounced in the subgroup of patients treated with parenteral nutrition since they started from higher values at the beginning of follow-up. 

## 4. Discussion

This study assessed the CP and the CA ratios in a population with a high prevalence of DRM and inflammation. The results suggest that both the CP ratio and the CA ratio are poor prognostic markers for complications during hospitalization, readmission, and LOS but slightly better for mortality. A CP ratio ≥ 0.153 and a CA ratio ≥ 0.040 had high sensitivity and NPV for mortality prediction, and both were predictors of mortality independently of age, comorbidities, cause of admission, and nutritional status. In addition, both ratios had poor accuracy in diagnosing DRM but may have a role in identifying severe DRM.

Other studies have previously evaluated the potential role of the CP ratio in predicting mortality. Most of them agree that a higher CP ratio is a predictor for risk of mortality. Yamada et al. reported that the optimal cut-off value for predicting mortality in patients hospitalized for acute heart failure was a CP ratio > 0.16 [7], quite similar to the cut-off found in this research. Another research showed that a CP ratio > 0.24 predicted mortality in patients admitted to the Intensive Care Unit (ICU) [19]. Also, the CP ratio has been reported as a predictor for overall survival and recurrence-free survival in patients with esophageal cancer [10]. 

The CA ratio has also been assessed as a predictor of mortality and previous studies have reported an optimal cut-off value of the CA ratio for this purpose quite close to that found in the current research. A CA ratio > 0.033 was related to lower overall survival in patients with cholangiocarcinoma [20], and a CA ratio > 0.036 predicted lower survival in patients with esophageal cancer [10]. Besides, different meta-analyses have found an association between a higher CA ratio and increased risk of mortality in patients with colorectal cancer [21], and hepatocellular cancer [22]. The cut-off value of the CA ratio for predicting mortality in the different studies included in those meta-analyses ranged from 0.026 to 0.671 [21,22]. In addition, other studies showed that the CA ratio is a predictor of mortality in non-neoplastic disorders, such as patients admitted for COVID-19 [23], acute pancreatitis [24], or acute heart failure [25].

The current research shows that both the CP ratio and the CA ratio are poor predictors for total, infectious, and non-infectious complications during hospitalization. Only the proportion of non-infectious complications was significantly higher in the subgroup of CA ratio ≥ 0.040. Nevertheless, the AUC reveals the CA ratio as a poor accurate predictor of these complications. Few studies have evaluated the role of these parameters as prognostic markers for complications, but most of them present results that are inconsistent with the present work. Llop-Talaveron et al. reported that the CP ratio was a predictor of sepsis in hospitalized patients treated with parenteral nutrition, while the CA ratio predicted infections and sepsis [11]. Other studies have also shown that the CA ratio is a predictor of postoperative complications after gastric cancer surgery [26], or for anastomotic leakage after colorectal surgery [27]. However, in this last study, the CA ratio had a low ability to detect complications other than anastomotic leakage [27]. The CP ratio has also been shown to be a predictor of major adverse cardiovascular events (MACE) after myocardial infarction [28]. The differences between the present results and the previous studies may be due partly to the heterogeneity of the complications collected in the current study. In contrast, the preceding investigations found an association between the CP ratio or the CA ratio and the occurrence of complications, mainly when analyzing one or a few specific complications. 

Likewise, only a few studies have previously evaluated the correlation between these ratios and the LOS. Li et al. found that a CP ratio > 0.24 was associated with a longer LOS in patients admitted to the ICU [19]. Other studies have shown that a higher CA ratio predicts a longer LOS in patients hospitalized for acute heart failure [25] or after gastric cancer surgery [26]. As a possible explanation for the discrepancy between the present results and the previous studies, it was hypothesized that there could be some bias in our research induced by the heterogeneity in the cause of hospital admission that altered the correlation between the CP and CA ratios and the LOS. However, this was ruled out since, when performing a stratified analysis by cause of admission, no correlation was found between the CP and CA ratios and the LOS in any of the subgroups.

On the other hand, as far as the authors know, no previous studies have assessed the ability of the CP ratio or the CA ratio to diagnose DRM or the degree of malnutrition. Therefore, the present results in this regard cannot be contrasted with the previous literature. It is striking that patients with severe DRM tend to have lower values of CP and CA ratios while, on the contrary, higher values of both ratios are related to the worst prognosis. At first, it was hypothesized that this could be attributed to the non-significant lower proportion of severe DRM in the subgroup of surgical patients, who presented higher values of both CP and CA ratios compared to the non-surgical patients. However, when analyzing the subgroups of surgical and non-surgical patients separately, the patients with severe DRM had lower values for their CP and CA ratios compared to those with moderate DRM or without DRM. A possible explanation for the lower CP and CA ratios in patients with severe DRM is that the inflammatory response could be partially compromised in those patients. Previous studies have shown that the increase in acute phase proteins in response to an infection is compromised in severely malnourished patients [29,30,31,32]. The impairment in the inflammatory response would possibly appear only in a severe degree of DRM, which may explain the results observed in this research. 

This study has some limitations. The wide heterogeneity in the patients included could have induced some bias that would explain why both the ability of the CP and the CA ratios to predict adverse outcomes and the specificity of their optimal cut-off points for mortality are lower than in other studies. Also, there is considerable variability regarding the time of the first determination of the CP and CP ratios, which coincides with the moment of the first nutritional assessment but was not necessarily performed in all patients at the same time after the admission to the hospital. On the other hand, the diversity regarding the cause of admission, comorbidities, and nutritional diagnosis among the patients included in the present study allows the results and cut-off points of the CP and the CA ratios to be extrapolated to any hospitalized patients. In addition, unlike previous studies that focus only on the ability of these ratios as prognostic markers, assuming that they reflect both the inflammation and nutritional status, this is the first research to explore the correlation between the CP and CA ratios and nutritional diagnosis. Therefore, based on the results of this study, the values of the CP ratio or the CA ratio could perhaps be used as an initial screening to detect severe DRM. They are easy-to-measure biomarkers and, even though their diagnostic accuracy is limited, they are superior to prealbumin and albumin for that purpose. Furthermore, the possible early detection of mortality risk based on the initial values of these biomarkers and their evolution during follow-up could facilitate prioritizing the initiation of nutritional therapy in these patients, as well as other procedures that can reduce the risk of mortality. 

## 5. Conclusions

The results of this study suggest that a CP ratio ≥ 0.153 or a CA ratio ≥ 0.040 are predictors of mortality. Both ratios have poor accuracy in diagnosing DRM, but a CP ratio < 0.237 or a CA ratio < 0.237 may help detect severe DRM. The CP ratio and the CA ratio improved the ability of the CRP to predict mortality, as well as the ability of prealbumin and albumin to detect severe DRM. Even so, the accuracy of both ratios as isolated markers is quite limited and they should be interpreted in combination with other clinical and laboratory markers. More studies are needed to clarify the role of the CP ratio and the CA ratio as prognostic markers and, in particular, as diagnostic markers of DRM and to establish optimal cut-off points. 

## Figures and Tables

**Figure 1 nutrients-16-02610-f001:**
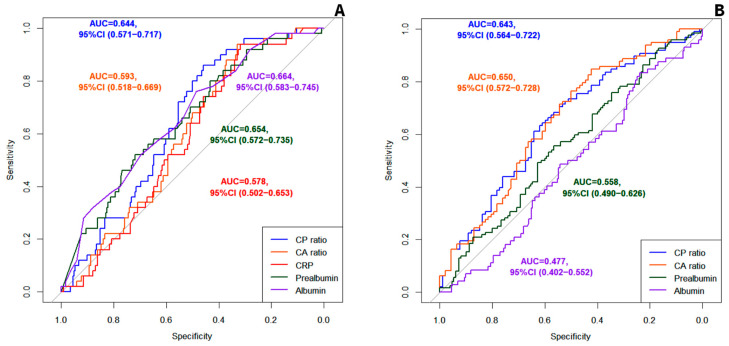
(**A**) ROC curves for mortality prediction. (**B**) ROC curves for differential diagnosis between severe and moderate disease-related malnutrition according to GLIM. AUC: Area under the curve. CP ratio: C-reactive protein to prealbumin ratio. CA ratio: C-reactive protein-to-albumin ratio, CRP: C-reactive protein.

**Figure 2 nutrients-16-02610-f002:**
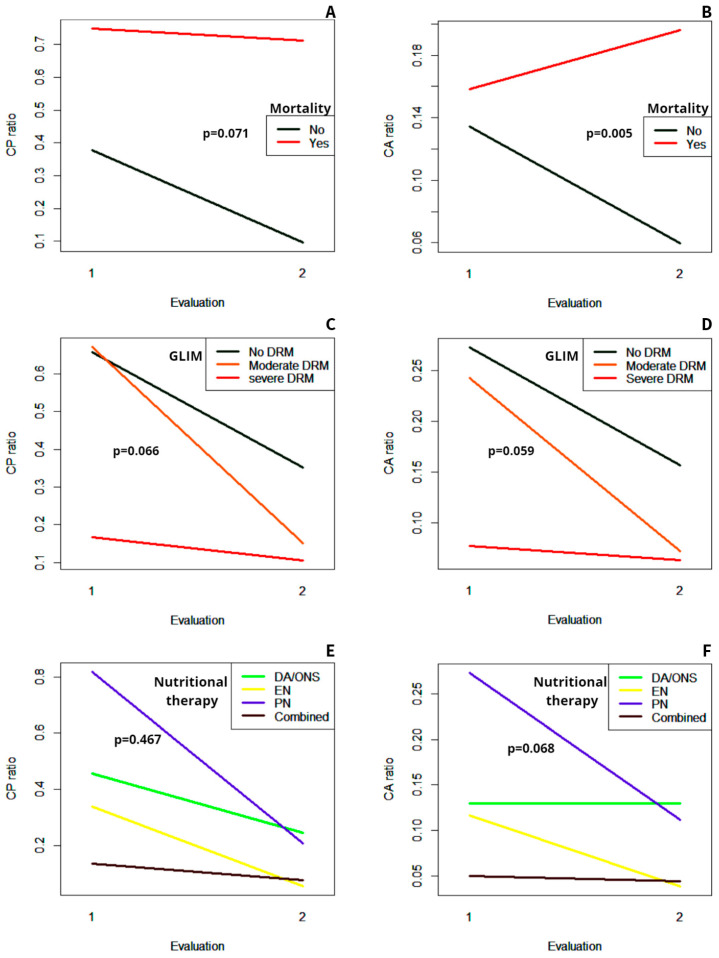
Evolution of the median of the CP ratio and the CA ratio between the first and the last assessment in the different subgroups of patients. (**A**) Evolution of the CP ratio in the patients who ended up dying versus those who did not. (**B**) Evolution of the CA ratio in the patients who ended up dying versus those who did not. (**C**) Evolution of the CP ratio in the different subgroups according to the nutritional diagnosis. (**D**) Evolution of the CA ratio in the different subgroups according to the nutritional diagnosis. (**E**) Evolution of the CP ratio in the different subgroups according to the type of nutritional therapy. (**F**) Evolution of the CA ratio in the different subgroups according to the type of nutritional therapy. GLIM: Global Leadership Initiative on Malnutrition; DRM: disease-related malnutrition; DA: dietary adjustment; ONS: oral nutritional supplements; EN: enteral nutrition; PN: parenteral nutrition.

**Table 1 nutrients-16-02610-t001:** Characteristics of the total patients and the subgroups based on the CP ratio and the CA ratio at first evaluation.

		Total Patients (*n* = 274)	CP Ratio < 0.153 (*n* = 110)	CP Ratio ≥ 0.153 (*n* = 164)	*p*	CA Ratio < 0.040 (*n* = 76)	CA Ratio ≥ 0.040 (*n* = 198)	*p*
Age		65 (SD 17)	64 (SD 18)	66 (SD 16)	0.303	63 (SD 18)	67 (SD 16)	0.083
Sex	Male	171 (62.4%)	70 (63.6%)	101 (61.6%)	0.731	46 (60.5%)	125 (63.1%)	0.690
	Female	103 (37.6%)	40 (36.4%)	63 (38.4%)		30 (39.5%)	73 (36.9%)	
Medical history	Diabetes mellitus	59 (21.5%)	22 (20%)	37 (22.6%)	0.613	15 (19.7%)	44 (22.2%)	0.654
	Hypertension	117 (42.7%)	46 (41.8%)	71 (43.3%	0.809	31 (40.8%)	86 (43.4%)	0.692
	Dyslipidemia	86 (31.4%)	33 (30%)	53 (32.3%)	0.685	23 (30.3%)	63 (31.8%)	0.804
	Obesity	28 (10.2%)	5 (4.5%)	23 (14%)	0.011	5 (6.6%)	23 (11.6%)	0.218
	IC	19 (6.9%)	7 (6.4%)	12 (7.3%)	0.886	5 (6.6%)	14 (7.1%)	0.886
	CHF	26 (9.5%)	12 (10.9%)	14 (8.5%)	0.551	9 (11.8%)	17 (8.6%)	0.410
	CKD	20 (7.3%)	11 (10%)	9 (5.5%)	0.159	8 (10.5%)	12 (6.1%)	0.203
	Liver disease	17 (6.2%)	6 (3.4%)	11 (6.7%)	0.673	5 (6.6%)	12 (6.1%)	0.873
	COPD	32 (11.7%)	11 (10%)	21 (12.8%	0.479	8 (10.5%)	24 (12.1%)	0.713
	OSA	15 (5.5%)	5 (4.5%)	10 (6.1%)	0.580	4 (5.3%)	11 (5.6%)	0.924
	Cancer	134 (48.9%)	49 (44.5%)	85 (51.8%)	0.273	31 (40.8%)	103 (52%)	0.096
Cause of admission	Surgery	127 (46.4%)	42 (38.2%)	85 (51.8%)	0.083	22 (28.9%)	105 (53%)	0.011
	Gastrointestinal disease	33 (12%)	13 (11.8%)	20 (12.2%)		12 (15.8%)	21 (10.6%)	
	Complications of cancer	29 (10.6%)	11 (10%)	18 (11%)		10 (13.2%)	19 (9.6%)	
	Neurologic disease	33 (12%)	19 (17.3%)	14 (8.5%)		12 (15.8%)	21 (10.6%)	
	Others	52 (19%)	25 (22.7%)	27 (16.5%)		20 (26.3%)	32 (16.2%)	
CONUT screening	No risk	126 (47.5%)	69 (64.5%)	57 (36.1%)	<0.001	49 (67.1%)	77 (40.1%)	0.001
	Moderate risk	116 (43.8%)	33 (30.8%)	83 (52.5%)		19 (26%)	97 (50.5%)	
	High risk	23 (8.7%)	5 (4.7%)	18 (11.4%)		5 (6.8%)	18 (9.4%)	
SGA	A	71 (25.9%)	25 (22.7%)	46 (28%)	0.013	19 (25%)	52 (26.3%)	0.202
	B	123 (44.9%)	42 (38.2%)	81 (49.4%)		29 (38.2%)	94 (47.5%)	
	C	80 (29.2%)	43 (39.1%)	37 (22.6%)		28 (36.8%)	52 (26.3%)	
DRM according to GLIM	No	84 (30.7%)	29 (26.4%)	55 (33.5%)	0.023	23 (30.3%)	61 (30.8%)	0.185
	Moderate	92 (33.6%)	31 (28.2%)	61 (37.2%)		20 (26.3%)	72 (36.4%)	
	Severe	98 (35.8%)	50 (45.4%)	48 (29.3%)		33 (43.4%)	65 (32.8%)	
GLIM phenotypic criteria ^†^	No criteria	83 (30.3%)	28 (25.5%)	55 (33.5%)	0.010	22 (28.9%)	61 (30.8%)	0.065
	1	18 (6.6%)	5 (4.5%)	13 (7.9%)		4 (5.3%)	14 (7.1%)	
	2	4 (1.4%)	0	4 (2.4%)		0	4 (2%)	
	3	32 (11.7%)	8 (7.3%)	24 (14.6%)		7 (9.2%)	25 (12.6%)	
	1 + 2	12 (4.4%)	4 (3.6%)	8 (4.9%)		1 (1.3%)	11 (5.6%)	
	1 + 3	60 (21.9%)	27 (24.5%)	33 (20.1%)		16 (21.1%)	44 (22.2%)	
	2 + 3	9 (3.3%)	6 (5.5%)	3 (1.8%)		6 (7.9%)	3 (1.5%)	
	1 + 2 + 3	56 (20.4%)	32 (29.1%)	24 (14.6%)		20 (26.3%)	36 (18.2%)	
GLIM etiologic criteria ^††^	No criteria	9 (3.3%)	8 (7.3%)	1 (0.6%)	<0.001	8 (10.5%)	1 (0.5%)	<0.001
	1	39 (14.2%)	33 (30%)	6 (3.7%)		32 (42.1%)	7 (3.5%)	
	2	29 (10.6%)	10 (9.1%)	19 (11.6%)		5 (6.6%)	24 (12.1%)	
	1 + 2	197 (71.9%)	59 (53.6%)	138 (84.1%)		31 (40.8%)	166 (83.8%)	
LOS		20 (IQR 26)	21 (IQR 32)	20(IQR 19)	0.970	20 (IQR 33)	20.5 (IQR 23)	0.729
Complications	Total	166 (60.6%)	65 (59.1%)	101 (61.6%)	0.679	41 (53.9%)	125 (63.1%)	0.164
	Infectious	111 (40.5%)	43 (39.1%)	68 (41.5%)	0.695	31 (40.8%)	80 (40.4%)	0.954
	Non-infectious	124 (45.3%)	47 (42.7%)	77 (47%)	0.491	27 (35.5%)	97 (49%)	0.045
Mortality		50 (18.2%)	7 (6.4%)	43 (26.2%)	<0.001	3 (3.9%)	47 (23.7%)	<0.001
Readmissions		72 (26.3%)	28 (25.5%)	44 (26.8%)	0.834	15 (19.7%)	57 (28.8%)	0.141

CP ratio: C-reactive protein-to-prealbumin ratio, CA ratio: C-reactive protein-to-albumin ratio, IC: Ischemic cardiomyopathy, CHF: chronic heart failure, CKD: chronic kidney disease, COPD: chronic obstructive pulmonary disease, OSA: obstructive sleep apnea, CONUT: CONtrolling NUTritional status, SGA: subjective global assessment, GLIM: Global Leadership Initiative on Malnutrition, DRM: disease-related malnutrition, LOS: length of stay, SD: standard deviation, IQR: interquartile range. ^†^ GLIM phenotypic criteria: 1 = weight loss >5% within past 6 months, 2 = low body mass index, and 3 = reduced muscle mass. ^††^ GLIM etiologic criteria: 1 = reduced food intake or assimilation and 2 = inflammation.

**Table 2 nutrients-16-02610-t002:** ROC curve analysis of the CP ratio and the CA ratio collected in the first evaluation to predict adverse outcomes and nutritional diagnosis.

	CP Ratio AUC	CA Ratio AUC	*p*
Mortality	0.644, 95%CI (0.571 to 0.717)	0.593, 95%CI (0.518 to 0.669)	0.029
Total complications	0.555, 95%CI (0.485 to 0.624)	0.559, 95%CI (0.489 to 0.629)	0.731
Infectious complications	0.539, 95%CI (0.468 to 0.610)	0.533, 95% CI (0.463 to 0.604)	0.720
Non-infectious complications	0.558, 95%CI (0.490 to 0.626)	0.561, 95%CI (0.493 to 0.629)	0.851
Readmission	0.529, 95% CI (0.452 to 0.605)	0.477, 95%CI (0.402 to 0.552)	0.499
SGA A (vs. B or C)	0.546, 95%CI (0.465 to 0.628)	0.572, 95%CI (0.489 to 0.655)	0.054
SGA B (vs. C)	0.614, 95% CI (0.534 to 0.695)	0.640, 95%CI (0.563 to 0.718)	0.136
SGA C (vs. A or B)	0.611, 95%CI (0.537 to 0.685)	0.643, 95% CI (0.573 to 0.712)	0.039
DRM according to GLIM (vs. no DRM)	0.535, 95% CI (0.460 to 0.610)	0.549, 95%CI (0.473 to 0.625)	0.375
Severe DRM (vs. moderate)	0.643, 95%CI (0.564 to 0.722)	0.650, 95%CI (0.572 to 0.728)	0.662

CP ratio: C-reactive protein-to-prealbumin ratio, CA ratio: C-reactive protein-to-albumin ratio, AUC: area under the curve, SGA: subjective global assessment, GLIM: Global Leadership Initiative on Malnutrition, DRM: disease-related malnutrition.

**Table 3 nutrients-16-02610-t003:** Prognostic and diagnostic accuracy of the optimal cut-off values of the CP ratio and the CA ratio.

		Sensitivity	Specificity	PPV	NPV
Prognostic accuracy for predicting mortality	CP ratio ≥ 0.153	0.86, 95%CI (0.73 to 0.94)	0.46, 95%CI (0.39 to 0.53)	0.26, 95%CI (0.20 to 0.34)	0.94, 95%CI (0.87 to 0.97)
	CA ratio ≥ 0.040	0.94, 95%CI (0.83 to 0.99)	0.33, 95%CI (0.26 to 0.39)	0.24, 95%CI (0.18 to 0.33)	0.96, 95%CI (0.89 to 0.99)
Diagnostic accuracy for identifying severe DRM	CP ratio < 0.237	0.61, 95%CI (0.51 to 0.71)	0.64, 95%CI (0.53 to 0.74)	0.65, 95%CI (0.54 to 0.74)	0.61, 95%CI (0.50 to 0.71)
	CA ratio < 0.273	0.85, 95%CI (0.76 to 0.91)	0.42, 95%CI (0.32 to 0.53)	0.61, 95%CI (0.52 to 0.69)	0.72, 95%CI (0.58 to 0.84).

CP ratio: C-reactive protein-to-prealbumin ratio; CA ratio: C-reactive protein-to-albumin ratio; PPV: positive predictive value; NPV: negative predictive value; DRM: disease-related malnutrition.

## Data Availability

The data presented in this study are available from the corresponding author upon request.

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
