# Peer review of "C-Reactive Protein-to-Prealbumin and C-Reactive Protein-to-Albumin Ratios as Nutritional and Prognostic Markers in Hospitalized Patients—An Observational Study"

_nutrients, 2024, doi:10.3390/nu16162610_

Round 1

Reviewer 1 Report

Comments and Suggestions for Authors

The findings of this study reveal that, while outperforming single protein indicators, the CP and CA ratios have a limited impact in predicting mortality and detecting severe DRM. However this study is interesting authors should address following comments:

1.      Page No. 1: Line No. 32 typo error nflammatory replace to inflammatory

2.      Are patients' levels of CRP and prealbumin assessed all at once, or at different times?

3.      The authors did check the fibrinogen  protein in the patients because it is another acute-phase reaction protein produced by the liver and is associated with inflammatory processes.

4.      The cut-off values for CRP-to-albumin and CRP-to-prealbumin ratios may vary from study to study. In this study, did the author notice any significant prognostic effects on the CRP-to-albumin and CRP-to-prealbumin ratios? please explain

Author Response

We would like to thank the reviewers for their critical reading of our work and their helpful and constructive corrections and suggestions. Following them, we have thoroughly revised the entire manuscript and modified the text, tables, and figures to incorporate the additional information and make the corrections requested by the reviewers. The modifications have been highlighted in the manuscript document. Below, we include our response to the Reviewer’s comments.

The findings of this study reveal that, while outperforming single protein indicators, the CP and CA ratios have a limited impact in predicting mortality and detecting severe DRM. However this study is interesting authors should address following comments:

  1. Page No. 1: Line No. 32 typo error nflammatory replace to inflammatory

We have corrected the error. Thank you for the observation.

  1. Are patients' levels of CRP and prealbumin assessed all at once, or at different times?

They were always collected at the same time, in the same blood test.

  1. The authors did check the fibrinogen  protein in the patients because it is another acute-phase reaction protein produced by the liver and is associated with inflammatory processes.

 No, it would have been interesting to evaluate this parameter as you say, but, unfortunately, we did not collect it.

  1. The cut-off values for CRP-to-albumin and CRP-to-prealbumin ratios may vary from study to study. In this study, did the author notice any significant prognostic effects on the CRP-to-albumin and CRP-to-prealbumin ratios? please explain

Yes, we observed a significant prognostic effect for mortality. Specifically, a CRP-to-albumin ratio ≥ 0.040 and/or a CRP-to-prealbumin ratio ≥ 0.153, were associated with a higher risk of mortality [OR = 11.38 95%CI (3.49 to 54.08), and OR = 7.98 95%CI (2.85 to 26.89), respectively], independently of the patient`s age, medical history, cause of admission, and nutritional diagnosis.  Nevertheless, the prognostic accuracy of both biomarkers is poor, as shown by the area under the ROC curve and the low specificity and positive predictive value of their cut-off values. Therefore, these ratios are biomarkers with a high sensitivity to identify the mortality risk. Still, they are not very specific and should be interpreted together with other clinical or laboratory data.

As you say, the cut-off points of these biomarkers vary between the different studies, but almost all of them agree that higher values of these ratios are associated with a worse prognosis.

Reviewer 2 Report

Comments and Suggestions for Authors

Reviewer’s Comments and Suggestions for Authors

Journal: Nutrients, MDPI

Manuscript ID: nutrients-3134834

Type: Article

Title: C-REACTIVE PROTEIN TO PREALBUMIN AND C-REACTIVE PROTEIN TO ALBUMIN RATIOS AS NUTRITIONAL AND PROGNOSTIC MARKERS IN OSPITALIZED PATIENTS

Authors: Rosa M. García-Moreno*, Laura Mola Reyes, Bricia López-Plaza and Samara Palma Milla

The authors of the Manuscript ID: nutrients-3134834 evaluated the role of the C-reactive protein to prealbumin (CP) ratio and the C-reactive protein to albumin (CA) ratio as nutritional and prognostic markers on hospitalized patients (n = 274). The results showed that  the CP and CA ratios showed greater accuracy in predicting mortality compared to C-reactive protein as an isolated marker (p = 0.011 and p = 0.006, respectively). Both ratios have poor accuracy in diagnosing DRM, but a CP ratio < 0.237 or a CA ratio < 0.237 may help detect severe DRM. The authors stated that the accuracy of both ratios as isolated markers is quite limited and they should be interpreted in combination with other clinical and laboratory markers.

The authors have described the main limitations of this study. The manuscript was presented too bad. Moreover, certain essential information regarding the experiments of this study was missing in the Materials and Methods section, which should be clarified. Therefore, this manuscript in its current form can not be recommended for publication.

Major revisions 

1. In the Materials and Methods section, certain essential information regarding the experiments of this study was missing. For example,

(1) In the “2.1. Study design and participants “, the information regarding the number of “ hospitalized patients older than 18 years” was missing. The similar issue as the basic information concerning the patients that were chosen in this study. Why did the authors to select these patients? Please clarify.

(2) The references or sources of the tools or standards used in this study should be cited, e.g., the CONtrolling NUTritional status (CONUT), Subjective Global Assessment (SGA) and the Global Leadership Initiative on Malnutrition (GLIM), and “The tool used for nutritional screening”.

(3) How did the authors to perform the albumin, prealbumin and high-sensitivity CRP assays? How to collect the samples? Please provide the detailed information.

(4) The authors should add additional subtitles to clarify the issues in the Materials and Methods section.

(5) Did “all determinations were measured” in triplicate?

2. In the Results section, for example:

(6)The authors should logically reconstruct this section as several subtitles, and highlight the major conclusions in each subtitle.

(7) The results in Table 1 should be analyzed in more details. Particularly, the characteristics of the subgroups of the patients should be described.

(8) The diversity regarding the cause of admission, comorbidities, and nutritional diagnosis among the patients included in this study should be analyzed.

(9) Figure 2: the same issue as Table 1; this Figure was not cited in the text; each of the small figures should be marked as A to F, and analyze individually in the text.

(10) Lines 184-185: Concerning the Finally, 128 patients had a second CP and CA ratios determination at the last assessment during hospitalization”, please provide more information of these patients.

Minor revisions

(11) Please format the Title according to the guidance of the journal for authors.

(12) Line 32: change to systemic inflammatory response”.

(13) Lines 58: “length of hospital stay (LOS)”, please check the abbreviation.

(14)Change the “(n=274)” to “(n = 274)” , and the p=0.002” to p = 0.002”. Please amend the similar writing issues throughout the article.

(15) Please format the Tables according to the guidance of the journal for authors.

Comments on the Quality of English Language

Moderate editing of English language required.

Author Response

We would like to thank the reviewers for their critical reading of our work and their helpful and constructive corrections and suggestions. Following them, we have thoroughly revised the entire manuscript and modified the text, tables, and figures to incorporate the additional information and make the corrections requested by the reviewers. The modifications have been highlighted in the manuscript document. Below, we include our response to the Reviewer’s comments.

Major revisions

  1. In the Materials and Methods section, certain essential information regarding the experiments of this study was missing. For example,

(1) In the “2.1. Study design and participants “, the information regarding the number of “ hospitalized patients older than 18 years” was missing. The similar issue as the basic information concerning the patients that were chosen in this study. Why did the authors to select these patients? Please clarify.

Among a total of 380 hospitalized patients assessed by the Nutrition Unit staff within the study period, 274 subjects met the selection criteria and were included. We have added this information in the Material and Methods section (lines 65-67).

(2) The references or sources of the tools or standards used in this study should be cited, e.g., the CONtrolling NUTritional status (CONUT), Subjective Global Assessment (SGA) and the Global Leadership Initiative on Malnutrition (GLIM), and “The tool used for nutritional screening”.

Thank you for the recommendation, we have cited the references.

(3) How did the authors to perform the albumin, prealbumin and high-sensitivity CRP assays? How to collect the samples? Please provide the detailed information.

We have included a subsection within the Material and Methods section explaining the collection of the blood samples and the laboratory measurement methods (lines 83-116).

(4) The authors should add additional subtitles to clarify the issues in the Materials and Methods section.

Additional subtitles have been added to the Material and Methods section. Thank you for the advice.

(5) Did “all determinations were measured” in triplicate?

No, they were not measured in triplicate. Regarding this, it should be taken into account that this is a study carried out in the context of our routine clinical practice.

  1. In the Results section, for example:

(6)The authors should logically reconstruct this section as several subtitles, and highlight the major conclusions in each subtitle.

Additional subtitles have been added to the Results section.

(7) The results in Table 1 should be analyzed in more details. Particularly, the characteristics of the subgroups of the patients should be described.

We have added a subsection within the Results section in which we analyzing and comparing the characteristics of these subgroups (lines 239-258).

(8) The diversity regarding the cause of admission, comorbidities, and nutritional diagnosis among the patients included in this study should be analyzed.

In regard to this issue, it should be considered that the study was carried out in routine clinical practice and that the selection criteria were not very restrictive. Logically, our population sample presents a high heterogeneity regarding these items since it reflects the patients admitted to our hospital. Because of the heterogeneity in the population characteristics, multivariate regression logistic analysis was performed and adjusted for all these variables (lines 207-212 and lines 216, 220). Anyway, the heterogeneity regarding the characteristics of the population sample has been included as a limitation of the study within the Discussion section (lines 381-384). 

(9) Figure 2: the same issue as Table 1; this Figure was not cited in the text; each of the small figures should be marked as A to F, and analyze individually in the text.

We have marked the small figures within Figure 2 as A-F. We have also included a new subsection within the Results section analyzing these figures by separate (lines 260-285).

(10) Lines 184-185: Concerning the “Finally, 128 patients had a second CP and CA ratios determination at the last assessment during hospitalization”, please provide more information of these patients.

In those patients whose hospital length of stay and follow-up by the Nutrition staff exceeded ten days, a second determination of these ratios was collected to assess the evolution of these parameters. We have included the information in the Material and Methods section (lines 131-134).

Minor revisions

(11) Please format the Title according to the guidance of the journal for authors.

The Title has been modified.

(12) Line 32: change to “systemic inflammatory response”.

We have changed it.

(13) Lines 58: “length of hospital stay (LOS)”, please check the abbreviation.

We have corrected it, thank you.

(14)Change the “(n=274)” to “(n = 274)” , and the “p=0.002” to “p = 0.002”. Please amend the similar writing issues throughout the article.

 We have revised the manuscript and corrected this.

(15) Please format the Tables according to the guidance of the journal for authors.

The Tables have been modified.

Comments on the Quality of English Language

Moderate editing of English language required.

We have revised and edited English language.

Reviewer 3 Report

Comments and Suggestions for Authors

The authors investigate an interesting question, the improved use of biomarkers to detect disease-related malnutrition at hospital admission. The results are fully presented, and the conclusion based on the results.

The following minor revisions could be considered:

title: I would include the aspect of malnutrition, e.g. markers "for malnutrition"

Line 82: revise sentence "it was collected data", e.g. data about ... was collected

My major comment would be about discussion and conclusion. The clinical aspects should be stressed, especially the nutritional aspects for readers of Nutrients. Can the results be used for triaging patients? Can the results be used to start a nutritonal therapy more early and improve the outcomes?

Line 273: should this be CP and CA rations?

Comments on the Quality of English Language

- the abstract should not be structured, remove, aims, results etc

- line 32: inflammatory

Author Response

We would like to thank the reviewers for their critical reading of our work and their helpful and constructive corrections and suggestions. Following them, we have thoroughly revised the entire manuscript and modified the text, tables, and figures to incorporate the additional information and make the corrections requested by the reviewers. The modifications have been highlighted in the manuscript document. Below, we include our response to the Reviewer’s comments.

Comments and Suggestions for Authors

The authors investigate an interesting question, the improved use of biomarkers to detect disease-related malnutrition at hospital admission. The results are fully presented, and the conclusion based on the results.

The following minor revisions could be considered:

title: I would include the aspect of malnutrition, e.g. markers "for malnutrition"

Thank you for the recommendation although, in our opinion,  “nutritional and prognostic markers” sounds better in the title. However, we have included it as a subtitle within the Results section: “Evaluation of the CP ratio and the CA ratio as predictors of malnutrition” (line 223).

Line 82: revise sentence "it was collected data", e.g. data about ... was collected

We have corrected this sentence (line 123-125).

My major comment would be about discussion and conclusion. The clinical aspects should be stressed, especially the nutritional aspects for readers of Nutrients. Can the results be used for triaging patients? Can the results be used to start a nutritonal therapy more early and improve the outcomes?

Thank you so much for the advice. We have added a paragraph at the end of the Discussion section emphasizing the potential clinical applications of these biomarkers (lines 394-400). 

Line 273: should this be CP and CA rations?

We have corrected the grammatical error.

Comments on the Quality of English Language

- the abstract should not be structured, remove, aims, results etc.

We have removed the subheading within the abstract.

- line 32: inflammatory

We have corrected the error.

Round 2

Reviewer 2 Report

Comments and Suggestions for Authors

Reviewer’s Comments and Suggestions for Authors

(Second round)

Journal: Nutrients, MDPI

Manuscript ID: nutrients-3134834

Type: Article

Title: C-REACTIVE PROTEIN TO PREALBUMIN AND C-REACTIVE PROTEIN TO ALBUMIN RATIOS AS NUTRITIONAL AND PROGNOSTIC MARKERS IN OSPITALIZED PATIENTS

Authors: Rosa M. García-Moreno*, Laura Mola Reyes, Bricia López-Plaza and Samara Palma Milla

The manuscript Manuscript ID: nutrients-3134834 has been amended according to most of my comments and suggestions for authors. However, there are still certain issues that should be clarified as below.

Essential revisions 

1. In the “2.3. Biochemical markers”, the authors should provide the source information of the reagents used in the assays, and the wave length for measuring the absorbance values.

2. Did “all determinations were measured” in triplicate?

The authors’ reply: No, they were not measured in triplicate. Regarding this, it should be taken into account that this is a study carried out in the context of our routine clinical practice.

The authors should state this in the Materials and Methods section.

3. The title should be written in lower-case letters, expect the first letter of each word in full.

4.Change the “(n=274)” to “(n = 274)” , and the “p=0.002” to “p = 0.002”. Please amend the similar writing issues throughout the article. The “n” and “pshould be written in italics.

5. Tables 1, 2, and 3: three horizontal and full lines of the tables are missing, please amend.

Comments on the Quality of English Language

Minor editing of English language required.

Author Response

We thank the reviewers for their critical reading of our work and helpful suggestions. Likewise, we apologize for having misunderstood some of the modifications suggested by Reviewer 2 in the previously submitted version of the manuscript. We have thoroughly revised the whole manuscript and incorporated the additional information and corrections requested by the reviewers. The modifications have been highlighted in the manuscript document. Below, we include our response to the Reviewer’s comments.

 The manuscript Manuscript ID: nutrients-3134834 has been amended according to most of my comments and suggestions for authors. However, there are still certain issues that should be clarified as below.

Essential revisions

  1. In the “2.3. Biochemical markers”, the authors should provide the source information of the reagents used in the assays, and the wave length for measuring the absorbance values.

The information about the reagents and the wavelengths has been added (lines 97-123).

  1. Did “all determinations were measured” in triplicate?

The authors’ reply: No, they were not measured in triplicate. Regarding this, it should be taken into account that this is a study carried out in the context of our routine clinical practice.

The authors should state this in the Materials and Methods section.

We have added this information in the Materials and Methods section (lines 87-88).

  1. The title should be written in lower-case letters, expect the first letter of each word in full.

The title has been modified following the instructions.

4.Change the “(n=274)” to “(n = 274)” , and the “p=0.002” to “p = 0.002”. Please amend the similar writing issues throughout the article. The “n” and “p” should be written in italics.

We have changed all of them.

  1. Tables 1, 2, and 3: three horizontal and full lines of the tables are missing, please amend.

The tables have been modified.

Comments on the Quality of English Language

Minor editing of English language required.

We have revised and edited English Language.